# Mixed-Lineage Leukaemia Gene Regulates Glucose-Sensitive Gene Expression and Insulin Secretion in Pancreatic Beta Cells

**DOI:** 10.3390/ijms25094704

**Published:** 2024-04-26

**Authors:** Satoshi Yoshino, Emi Ishida, Kazuhiko Horiguchi, Shunichi Matsumoto, Yasuyo Nakajima, Atsushi Ozawa, Masanobu Yamada, Eijiro Yamada

**Affiliations:** Department of Internal Medicine, Division of Endocrinology and Metabolism, Gunma University Graduate School of Medicine, Maebashi 371-8511, Japan; syoshino@gunma-u.ac.jp (S.Y.); k-hori@gunma-u.ac.jp (K.H.); smatsu@gunma-u.ac.jp (S.M.); yasuyonakajima11@gmail.com (Y.N.); ozawaa@gunma-u.ac.jp (A.O.); myamada@gumna-u.ac.jp (M.Y.)

**Keywords:** diabetes, beta cells, glucose responsiveness, MLL, SLC2A1, SLC2A2, insulin

## Abstract

The escalating prevalence of diabetes mellitus underscores the need for a comprehensive understanding of pancreatic beta cell function. Interest in glucose effectiveness has prompted the exploration of novel regulatory factors. The myeloid/lymphoid or mixed-lineage leukaemia gene (*MLL*) is widely recognised for its role in leukemogenesis and nuclear regulatory mechanisms through its histone methyltransferase activity in active chromatin. However, its function within pancreatic endocrine tissues remains elusive. Herein, we unveil a novel role of *MLL* in glucose metabolism and insulin secretion. *MLL* knockdown in βHC-9 pancreatic beta cells diminished insulin secretion in response to glucose loading, paralleled by the downregulation of the glucose-sensitive genes *SLC2a1* and *SLC2a2*. Similar observations were made in *MLL* heterozygous knockout mice (*MLL*+/−), which exhibited impaired glucose tolerance and reduced insulin secretion without morphological anomalies in pancreatic endocrine cells. The reduction in insulin secretion was independent of changes in beta cell mass or insulin granule morphology, suggesting the regulatory role of *MLL* in glucose-sensitive gene expression. The current results suggest that MLL interacts with circadian-related complexes to modulate the expression of glucose transporter genes, thereby regulating glucose sensing and insulin secretion. Our findings shed light on insulin secretion control, providing potential avenues for therapeutics against diabetes.

## 1. Introduction

The prevalence of diabetes mellitus is steadily rising, underscoring the pressing need for a comprehensive understanding of pancreatic beta cell function. Given that insulin secretion governs blood glucose levels, elucidating the intricate mechanisms underlying insulin secretion mechanisms is imperative. Recent scholarly discourse has reignited interest in the concept of glucose effectiveness [1].

The myeloid/lymphoid or mixed-lineage leukaemia (*MLL*) gene, also known as *KMT2A* (lysine methyltransferase 2A) based on the function of the protein it encodes, is frequently translocated to the long arm q23 region (11q23) of chromosome 11 in blood malignancies. This translocation results in an *MLL* fusion gene, which drives leukemogenesis [2]. Indeed, *MLL* functions in a manner similar to that of the Drosophila trithorax (*Trx*) gene, crucial for nuclear regulatory mechanisms. It establishes an epigenetic transcriptional memory system via SET domain-dependent histone methyltransferase activity. Specifically, *MLL* targets a lysine 4 residue within histone 3 (H3K4), which is primarily associated with active chromatin regions [3,4,5]. MLL1, a member of the MLL family, is crucial for maintaining specific Hox gene expression during morphogenesis. It is a component of a vast chromatin remodelling complex in which menin plays a pivotal role, along with other key regulators such as WD repeat domain 5, ASH2-like, histone lysine methyltransferase complex subunit, RB-binding protein 5, and histone lysine methyltransferase complex subunit [5,6,7]. These are also part of alternative nuclear complexes, suggesting coordinated chromatin remodelling. Notably, two independent studies in 2004 concurrently demonstrated the association of MLL with menin within a nuclear protein complex, facilitating the transcriptional activation of Hox genes [8,9]. More recently, the collaborative regulation of cyclin-dependent kinase inhibitors (CDKIs), notably p27Kip1/CDKN1B and p18Ink4C/CDKN2C, by MLL and menin has been revealed. Further, some mutations in menin can impair the stimulation of these CDKI genes, consequently promoting aberrant cell proliferation [10,11].

We previously demonstrated the haploinsufficiency and predominant expression of multiple endocrine neoplasia type 1 (MEN1)-related genes, including MLL, p27Kip1, and p18Ink4C, across various endocrine organs [12]. Furthermore, MLL can target histones within metabolic gene loci as a methyltransferase [13]. Nonetheless, the functional significance of MLL within pancreatic endocrine tissues remains unclear. Therefore, in this study, we aimed to elucidate the role of MLL in glucose metabolism, particularly focusing on its involvement in insulin secretion mechanisms.

## 2. Results

### 2.1. Insulin Secretion Decreased in Response to Glucose Loading in βHC-9 Cells

To investigate the functions of MLL in insulin secretion by pancreatic beta cells, we employed the βHC-9 cell line. The *Kmt2a* gene, which encodes MLL, was knocked down using small interfering RNA (siRNA) in these cells. As shown in Figure 1A, approximately 80% of *Kmt2a* mRNA expression was silenced following transfection. *MLL*-knockdown βHC-9 cells were then subjected to glucose-stimulated insulin secretion (GSIS) tests. As shown in Figure 1B, *MLL* knockdown decreased insulin secretion (insulin levels corrected for mg protein) in response to a glucose challenge. Insulin secretion was suppressed by approximately 20% upon a glucose stimulus of 5.6 mM and by approximately 40% upon a high glucose stimulus of 11.2 mM. To elucidate the mechanism underlying the decreased insulin secretion, we explored gene expression changes via microarray analysis. As shown in Figure 1C, *MLL*-knockdown cells exhibited 1439 downregulated and 1774 upregulated genes compared with controls. Approximately 500 genes were upregulated more than two-fold, with a similar number of genes exhibiting a corresponding decrease relative to control cells. Glucose-sensitive SLC2a1 (glucose transporter type 1, GLUT1) and SLC2a2 (glucose transporter type 2, GLUT2) were the most differentially regulated genes, with both being significantly downregulated following *MLL* knockdown (Figure 1D). These results suggest that MLL mediates the expression of glucose-sensitive genes and promotes insulin secretion in response to glucose loading in vitro.

### 2.2. Metabolic Profile of MLL Heterozygous Knockout Mice

Although *MLL* mutant mice have been extensively generated, *MLL* deficiency is embryonic lethal [14,15,16]. Reports have suggested that *MLL*+/− mice exhibit mild anaemia compared with *MLL*+/+ littermates. However, no difference in leukocyte count has been noted [16].

As the metabolic profile of *MLL* heterozygous knockout mice (*MLL*+/−) remains unexplored, we determined their body mass, dietary consumption, and lipid profile before analysing their glucose metabolism. As shown in Figure 2A, *MLL*+/− mice exhibited considerable weight reduction compared with their wild-type (WT) counterparts, consistent with prior observations. However, no discernible disparities were observed in food intake or lipid profile (Figure 2B,C).

### 2.3. Insulin Secretion Decreased in MLL Heterozygous Knockout Mice

To validate the GSIS test results in *MLL*-knockdown βHC-9 cells, we investigated glucose metabolism in 20-week-old *MLL*+/− mice. Intraperitoneal glucose tolerance tests (IPGTTs) revealed compromised glucose tolerance in *MLL*+/− mice compared with WT mice (Figure 3A). Meanwhile, insulin tolerance tests (ITTs) indicated no insulin resistance and diminished insulin secretion upon glucose challenge compared with WT mice (Figure 3B,C). These data suggest that *MLL*+/− mice exhibit impaired glucose tolerance, with reduced early-phase insulin secretion, corroborating the GSIS results in *MLL*-knockdown βHC-9 cells and underscoring the substantial involvement of MLL in insulin secretion.

### 2.4. Impaired Insulin Secretion and Downregulation of Glucose-Sensitive Genes in MLL Heterozygous Knockout Mice

Next, we investigated changes in the islets and beta cells of the model mice. Islets were meticulously harvested from mouse pancreatic tissues, and GSIS tests were performed. In agreement with the GSIS results from the cell lines, insulin secretion decreased in the islets from the pancreata of *MLL*+/− mice in response to glucose challenge compared with those from WT mice (Figure 4A). More importantly, in line with our observations in *MLL*-knockdown βHC-9 cells, *SLC2A1* and *SLC2A2* mRNA were downregulated in the islets from *MLL*+/− mice compared with those from WT mice (Figure 4B). Subsequently, histological examination was performed to ascertain whether the reduced insulin secretion stemmed from morphological or histological anomalies. Immunohistochemical staining revealed similar staining patterns for both insulin and glucagon within the pancreata of *MLL*+/− and WT mice. Moreover, no discernible disparities were noted in the quantity or morphology of pancreatic beta cells between the two groups (Figure 4C,D). More intriguingly, electron microscopy revealed no disparities in the abundance or morphology of insulin secretory granules and other subcellular structures between the pancreata of *MLL*+/− and WT mice (Figure 4E).

These findings indicate that the reduced insulin secretion from islets following glucose challenge in *MLL*+/− mice is not attributable to a decrease in the number of beta cells, depletion of insulin-secreting granules, or morphological aberrations within the islets. Instead, *MLL* haploinsufficiency may attenuate insulin secretion in response to glucose stimulation by downregulating the expression of glucose-sensitive genes.

### 2.5. MLL Heterozygous Mouse Islets Did Not Exhibit Apoptosis

*MLL*+/− mice exhibited no pancreatic histological anomalies or reductions in insulin content or insulin-secreting cell numbers. MLL, together with Men1, plays crucial roles in the development of endocrine tumours by modulating the p27Kip1/CDKN1B (p27) and p18Ink4C/CDKN2C (p18) axes [10,11,12]. Moreover, Men1 is responsible for multiple endocrine neoplasia type 1 (MEN1), a hereditary disorder causing tumours in several endocrine glands, including the pancreas as well as the parathyroid and pituitary glands [17,18]. Thus, we analysed tumour-related genes and apoptosis within the islets and observed no discernible differences in the expression of menin, P18, and P27 in the islets of *MLL*+/− mice (Figure 5A). TUNEL staining indicated no increase in apoptosis within the pancreatic islets of *MLL*+/− mice (Figure 5B). These findings reinforce the notion that MLL does not regulate tumourigenesis or apoptosis within islets and that the reduced insulin secretion cannot be attributed to tumourigenesis or reductions in cell number.

## 3. Discussion

In the present study, we investigated the role of MLL in pancreatic endocrine tissues. We examined glucose metabolism and insulin secretion in *MLL*+/− mice and *MLL*-knockdown pancreatic cell lines. Suppressing MLL expression in both cell lines and mouse islets led to reduced insulin secretion in response to glucose stimulation. Further, we observed decreased expression of the glucose-sensitive genes *SLC2A1* and *SLC2A2*, known to be regulated by *MLL*. These findings suggest that *MLL* plays a crucial role in modulating glucose-sensitive gene expression and insulin secretion.

We observed diminished insulin secretion in response to glucose challenge in *MLL*-knockdown βHC-9 cells. Although βHC-9 cells, which are derived from mouse pancreatic islets, can secrete insulin in response to glucose in a concentration-dependent manner, they originate from the hyperplastic islets of transgenic mice expressing the simian virus 40 tumour antigen. Therefore, βHC-9 cells may exhibit characteristics different from those of beta cells in tissues in terms of tumourigenesis and apoptosis [19]. Despite extensive studies on its role in leukaemia, the contribution of MLL to glucose metabolism remains elusive [19,20,21,22,23]. In the present study, we subjected 20-week-old WT and MLL+/− mice to IPGTT and ITT (Figure 3). The blood insulin levels (Figure 3B) in the IPGTT may reflect insulin levels after glucose load, confirming insulin secretion rather than synthesis from the pancreas. In the context of insulin secretion, we hypothesised the presence of both basal and additional secretion, with each being possibly governed by distinct mechanisms. The similar fasting blood glucose levels between WT and MLL+/− mice suggested insulin resistance and no differences in basal insulin secretion. Furthermore, the ITT suggested comparable insulin resistance in both mouse types. Therefore, we inferred that early insulin secretion is decreased in MLL-knockout mice in response to glucose load; however, insulin resistance does not occur, resulting in similar blood glucose levels at 0 and 120 min. Based on these results, we concluded that MLL+/− mice exhibit impaired glucose tolerance and reduced insulin secretion, in line with our data from βHC-9 cells. Furthermore, *MLL*+/− mice display a diminutive physique, as previously reported [14]. Interestingly, no differences in food intake were noted in MLL+/− mice, with no discernible developmental anomalies, suggesting that this phenotype may stem from diminished insulin secretion.

Considering the involvement of MLL in tumourigenesis and the reduced insulin secretion in *MLL*+/− mice, we performed a comprehensive histological analysis of pancreatic sections. However, no aberrations in cellular count or morphology that would reflect tumour formation or increased apoptosis were noted in 20-week-old mice. Furthermore, no disparities were observed with respect to insulin and glucagon staining. Moreover, electron microscopy revealed no differences in insulin granule abundance. Tumour-related gene expression was not differentially regulated in 20-week-old *MLL*+/− mice. These findings posit that MLL does not affect the morphology, tumourigenesis, or insulin content of pancreatic endocrine cells in middle-aged mice. Menin has been reported to suppress the expression and secretion of insulin in an insulinoma cell line [24]. Nevertheless, no discernible change in menin expression was observed in the pancreas of *MLL*+/− mice. Furthermore, inhibition of menin–MLL and TGF-β synergistically promotes the proliferation of human beta cells but does not correlate with diminished INS expression [25]. Despite confirming the absence of changes in insulin gene expression and protein levels, insulin secretion in response to glucose stimulation was not assessed. The authors assert that this cannot be indicative of the overall insulin secretion from the beta cell populace or changes in beta cell count. Moreover, MLL-independent insulin secretion was not addressed and therefore remains ambiguous. Others have reported that small-molecule inhibition of TGF-β did not affect GSIS in primary human beta cells [26]. A comprehensive analysis of our findings and prior reports suggests that only MLL may be implicated in insulin secretion in response to glucose stimulation. 

In this study, we demonstrated the involvement of MLL in insulin secretion upon glucose stimulation. Regarding the underlying mechanism, we observed the downregulation of SLC2A1 and SLC2A2 in the islets of *MLL*+/− mice and *MLL*-knockdown βHC9 cells. *SLC2A2* encodes GLUT2, originally named HepG2/erythrocyte/brain transporter, owing to its expression on red blood cells [27,28,29]. GLUT2 is expressed in various tissues, including the liver, intestine, kidneys, and central nervous system. GLUT2 predominantly participates in glucose uptake in hepatocytes and renal tubular epithelial cells. Therefore, fluctuations in blood glucose levels during the IPGTT may have been affected. Although GLUT2 is also expressed in small intestine epithelial cells, we assumed that minimal glucose passes through the intestine during the IPGTT; therefore, it did not affect our investigation. In the central nervous system, GLUT2 is implicated in hypoglycaemic responses, although we assumed it had no effect in this in vivo study [30]. However, GLUT2 is primarily expressed in pancreatic islet beta cells, underscoring its pivotal role in glucose sensing [31,32,33,34]. *SLC2A1* encodes GLUT1, which is predominantly expressed at the blood–brain barrier and facilitates the energy-independent transport of glucose into the brain. Furthermore, GLUT1 is expressed in the liver, kidneys, and small intestine. Therefore, when considering the liver and kidneys, its involvement in the IPGTT results, similar to GLUT2, cannot be dismissed [27,28]. However, SLC2A1 also exhibits higher expression in humans compared with mouse beta cells [31,32]. In fact, some studies suggest that GLUT1 is the principal glucose transporter in human insulin-secreting beta cells [31,32]. Furthermore, Ohtsubo et al. have demonstrated the involvement of both GLUT2 and GLUT1 in glucose uptake by human beta cells [33]. Research on rodent diabetes models and islets from patients with type 2 diabetes (T2D) suggests a correlation between the decrease in GLUT1 and GLUT2 expression and the development of T2D [35,36,37,38]. Given the roles of GLUT1 and GLUT2, it is postulated that the reduced expression of SLC2A1 and SLC2A2 in *MLL*-knockout mice contributes to diminished insulin secretion upon glucose stimulation.

How MLL regulates the expression of SLC2A1 and SLC2A2 is still under investigation. In this regard, Katada et al. reported the physical interaction between MLL1 and CLOCK at CLOCK-controlled promoters. They emphasised the critical role of MLL1 and H3K4 methylation in establishing a chromatin-permissive state necessary for circadian gene expression [13]. Disruption of CLOCK and BMAL1, components of the circadian clock, results in hypoinsulinemia [39]. Further, impaired GSIS has been reported in mice lacking Bmal1 [40]. The expression of Slc2a2 in rodent beta cells is rapidly regulated by glucose stimulation [41]. Furthermore, the complexity of GLUT2 regulation has been further elucidated by studies indicating circadian and age-dependent modulation of GLUT2 [42,43]. Taken together, it is plausible that the histone methyltransferase MLL interacts with circadian-related complexes to regulate accessibility at the promoter of the mouse SLC2A2 gene, facilitating expression. Although we did not determine this mechanism in the present work, research efforts in this direction are currently ongoing.

In this study, we investigated insulin secretion in mice, acknowledging potential physiological differences between humans and mice. However, in both species, the mechanisms underlying insulin secretion from the islets remain incompletely elucidated. MLL may be implicated in glucose sensitivity and insulin secretion in humans. The clinical implication of our study lies in its potential application to humans. If the same mechanisms can be confirmed, aspects of insulin secretion and glucose effectiveness may be elucidated, potentially contributing to the treatment of diabetes.

In summary, we have discerned that MLL attenuates insulin secretion by downregulating the expression of glucose-sensitive genes in mice. This finding provides novel insights into insulin secretion control and glucose effectiveness, which are of major relevance for diabetes treatment.

## 4. Materials and Methods

### 4.1. Animals and Islet Extraction

Procedures for animal care and use in this study were approved by the Review Committee on Animal Use at Gunma University, Maebashi, Japan. *MLL*+/− mice carrying the anti-OVA-specific TCR-ab (DO11.10) transgene were kindly provided by Prof. T. Nakayama (Chiba University, Chiba, Japan). *MLL*+/− mice were originally generated by Prof. T. Komori (Nagasaki University, Japan). Animals were maintained on a 12-h light/dark schedule (lights on at 6:00 h), with ad libitum access to laboratory chow and tap water. All experiments were performed between 11:00 and 13:00 h. Pancreatic islets were isolated from mice according to a previously described protocol [44]. In brief, pancreatic islets were digested with collagenase (Sigma #C7657, Tokyo, Japan) and vigorously dissociated by mechanical pipetting. Islets were then hand-picked under a dissecting microscope and pooled for further analysis. 

### 4.2. Biochemical Parameters and Insulin and Glucose Tolerance Tests

Twenty-week-old mice were subjected to IPGTTs and ITTs as previously reported [45]. The concentration of insulin was measured using an ELISA kit (Mercodia, Winston Salem, NC, USA).

### 4.3. Cell Culture

βHC-9 cells were kindly provided by Dr. Douglas Hanahan (University of California, San Francisco, CA, USA) and were maintained in Dulbecco’s modified Eagle medium supplemented with high glucose, 10% foetal bovine serum, 100 units/mL penicillin, 100 units/mL streptomycin, and 20 mM glutamine, under humidified conditions of 5% CO_2_ at 37 °C.

### 4.4. Knockdown of Kmt2A by siRNA

*Kmt2A* knockdown in βHC-9 cells was achieved by transfecting siRNA via electroporation, as previously reported [46]. Pooled siRNA oligonucleotides targeting *Kmt2A* were designed and synthesised at Dharmacon Research (Lafayette, CO, USA, siGENOME SMART pool M-040631-01-0010). Pooled unrelated siRNA (siCONTROL Non-Targeting siRNA pool, D-001206-13-20) was used as a control. 

### 4.5. RNA Extraction and Real-Time Polymerase Chain Reaction (PCR)

Total RNA was extracted from mouse islets and βHC-9 cells using the RNeasy Mini Kit (QIAGEN, Hilden, Germany) according to the manufacturer’s instructions. Real-time PCR was performed using TaqMan probes and the Applied Biosystems 7500 sequence detection system, as previously reported [45]. The TaqMan probes for Gapdh (Mm99999915_g1), Kmt2A (Mm01179235_m1), Slc2a1 (Mm05908127_s1), and Slc2a2 (Mm00446229_m1) were obtained from Applied Biosystems (Bedford, MA, USA).

### 4.6. Glucose-Stimulated Insulin Secretion (GSIS)

The islets, cultured overnight after isolation, were preincubated with a buffer containing 2.8 mM glucose (the buffer contained 118.5 mM NaCl, 2.54 mM CaCl_2_, 1.19 mM KH_2_PO_4_, 1.19 mM MgSO_4_, and 10 mM HEPES, BSA; pH 7.4) for 15 min and centrifuged (2500 rpm, 5 min). Then, the supernatant was replaced with buffer containing 2.8–16.7 mM glucose, followed by incubation for 60 min. The supernatant obtained via centrifugation (2500 rpm, 5 min) was stored at −20 °C until insulin concentration measurement via ELISA.

### 4.7. Histology and Electron Microscopy

Tissue sections were prepared and immunohistochemistry was performed as previously described [45]. Electron microscopy analysis was performed by Tokai Electron Microscopy Analysis Co., Ltd. (Nagoya, Japan).

### 4.8. Statistical Analyses

Statistical analyses were performed with analysis of variance (ANOVA) and Student’s *t*-test or the Wilcoxon/Kruskal–Wallis test using JMP (SAS Institute Inc., Cary, NC, USA).

## Figures and Tables

**Figure 1 ijms-25-04704-f001:**
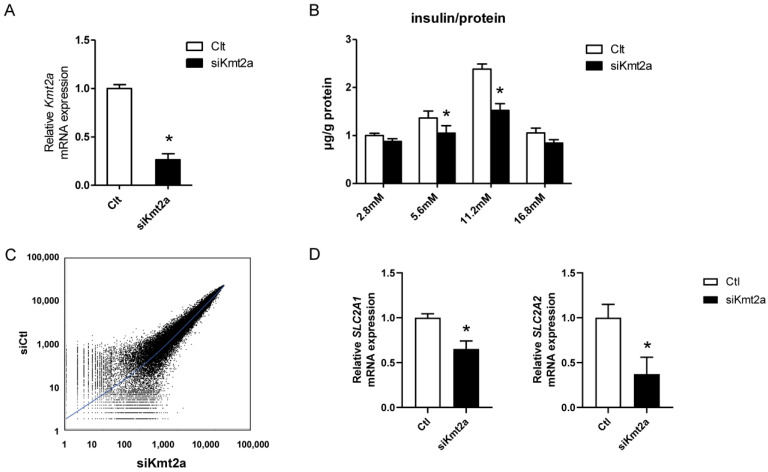
Decrease in insulin secretion in response to glucose loading in *Kmt2a*-knockdown βHC-9 cells. (**A**) Kmt2A expression was silenced via siRNA-mediated *Kmt2A* knockdown. Small interfering RNA targeting *Kmt2A* (siKmt2A) or control siRNA (Ctl) (200 nM) was introduced into βHC-9 cells via electroporation, and *Kmt2A* mRNA expression was quantified after 48 h via real-time polymerase chain reaction. (**B**) Glucose-stimulated insulin secretion tests were performed in *Kmt2a*-knockdown βHC-9 cells. (**C**) Scatter plot of the microarray data of *Kmt2A*-knockdown βHC-9 cells. (**D**) SLC2a1 and SLC2a2 expression. Data represent the mean ± standard error of mean from triplicate samples. Asterisks indicate significant differences from Ctl-transfected cells. The experiment was repeated twice, with similar results. The white and black columns represent the siClt- and siKmt2A-transfected groups, respectively.

**Figure 2 ijms-25-04704-f002:**
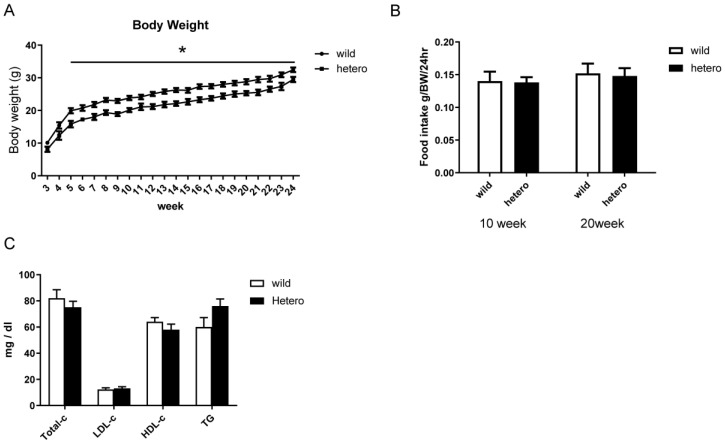
Metabolic profile of *MLL* heterozygous knockout mice. (**A**) Bodyweight (BW) gain (n = 30 in each group), (**B**) cumulative food intake (n = 30 in each group), and (**C**) lipid profile (n = 10 in each group) were assessed in wild-type (WT) and *MLL*+/− mice. Data are presented as the mean ± standard error of mean. Asterisks indicate significant differences from WT mice. White and black columns represent WT and *MLL*+/− mice, respectively.

**Figure 3 ijms-25-04704-f003:**
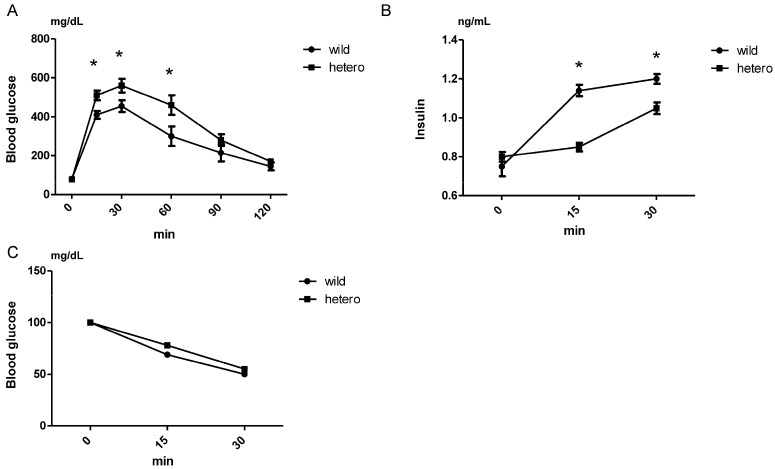
Decreased insulin secretion in heterozygous *MLL*-knockout mice. (**A**,**B**) Glucose intolerance in mice. WT and MLL+/− mice (20 weeks old) received an i.p. injection of glucose (1.0 g glucose/kg bw) for 20 weeks. Blood samples were collected at the indicated times, whereafter glucose (0, 30, 60, 90, and 120 min) (**A**) and insulin (0, 15, and 30 min) (**B**) levels were measured. Each group included 10 mice. (**C**) The insulin tolerance test (ITT) was performed in WT and *MLL*+/− mice via i.p. injection of regular insulin (0.5 U/kg bw) for 20 weeks. Subsequent blood samples were used to determine blood glucose levels. Data are expressed as relative changes from levels before insulin injection (n = 10 in each group). Data are expressed as mean ± standard error of mean. Asterisks indicate significant differences from WT mice.

**Figure 4 ijms-25-04704-f004:**
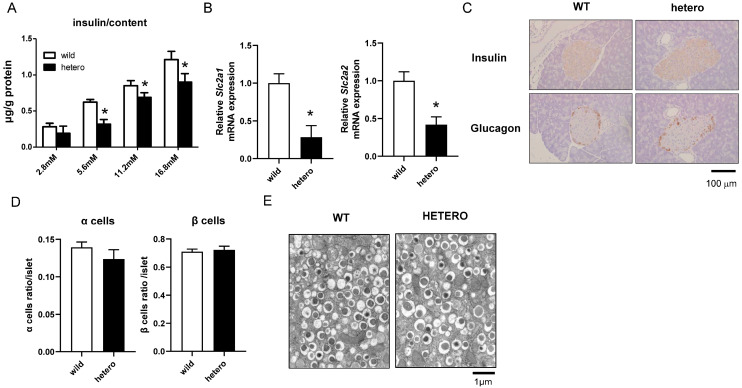
Impaired insulin secretion owing to the decreased expression of glucose-sensitive genes in *MLL* heterozygous knockout mice. (**A**) Glucose-stimulated insulin secretion (GSIS) tests were performed using islets purified from WT and *MLL*+/− mice. Data are expressed as the mean ± standard error of mean from triplicate samples. Asterisks indicate significant differences from WT mouse islets. The experiment was repeated twice, with similar results. The white and black columns represent the WT and *MLL*+/− mouse islets, respectively. (**B**) *SLC2a1* and *SLC2a2* expression was determined via real-time polymerase chain reaction. Data are expressed as the mean ± SEM from triplicate samples. Asterisks indicate significant differences from WT mouse islets. The experiment was repeated twice, with similar results. White and black columns represent the WT and *MLL*+/− mouse islets, respectively. (**C**,**D**) Immunohistochemistry for insulin and glucagon in the pancreas. Representative photos from WT (**left** panel) and *MLL*+/− mice (**right** panel). Scale bars indicate 100 μm. The number of alpha and beta cells in WT and *MLL*+/− mouse islets. Data are expressed as the mean ± standard error of mean from 15 islets. (**E**) Electron microscopy images of the islets. Photographs are representative of WT (**left** panel) and *MLL*+/− mice (**right** panel).

**Figure 5 ijms-25-04704-f005:**
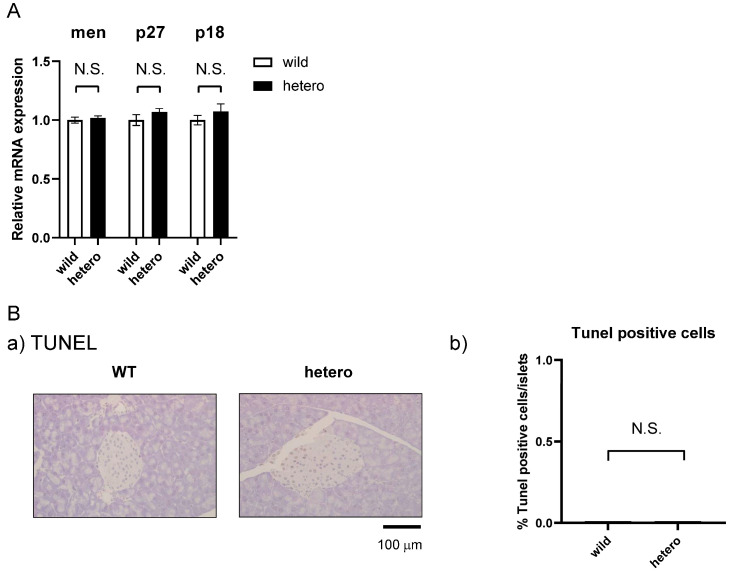
*MLL* heterozygous mouse islets did not exhibit increased apoptosis. (**A**) Menin, p27, and p18 mRNA expression was determined via real-time polymerase chain reaction. Data are expressed as the mean ± standard error of mean from triplicate samples. The experiment was repeated twice, with similar results. White and black columns represent the WT and *MLL*+/− mouse islets, respectively. (**B**) (**a**) TUNEL staining of the pancreas. Representative images of WT (**left** panel) and *MLL*+/− mice. Scale bars indicate 100 μm. (**b**) Number of TUNEL-positive cells in WT and MLL+/− mouse islets. Data are expressed as the mean ± standard error of mean from 15 islets.

## Data Availability

The original contributions presented in the study are included in the article, further inquiries can be directed to the corresponding author.

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
