# Peer review of "Mixed-Lineage Leukaemia Gene Regulates Glucose-Sensitive Gene Expression and Insulin Secretion in Pancreatic Beta Cells"

_ijms, 2024, doi:10.3390/ijms25094704_

Round 1
Reviewer 1 Report
Comments and Suggestions for Authors
Congratulations on the article. The authors present a pretty well conducted analysis of the role of mixed-lineage leukemia gene in the structure and functionality of pancreatic beta cells in mice models. Insulin secretion control remains the key stone of treatment for diabetes, particularly in diabetes mellitus type 2 when insulin leads to declined insulin production, and eventual pancreatic beta-cell failure, so there is value of reporting this kind of analysis from a pathophysiological and therapeutical point of view.
The article is well-written, and data is clearly presented. It contributes to the body knowledge of glucose homeostasis and insulin secretion.
There are very few points to be considered which would be beneficial for this paper.
Authors conclude that "MLL does not regulate tumorigenesis or apoptosis within islets", but it has been previously well-reported that MII deletion may induce leukemogenesis and fatal bone marrow failure occuring in a few weeks (in animal models). At what age were these KO mice sacrified? Did authors perform an assessment on haemathological features? Do authors consider the long-life period in mice before being sacrified enough to assure "MLL does not regulate tumorigenesis within islets"? It would be convenient a further explanation on histologycal analysis since refrence (44) do not fully explain this point.
Authors should also further discuss about results showed in Figure 3. How do they explain, the islets in heterozygous MLL-knockout mice begin to synthesize more insulin 30 minutes after exposure to i.p glucose? (and blood sugar levels decrease at a time until they reach the same blood levels than wild-tipe mice after 120 min).
It is also difficult to understand why differences in glucose homeostasis in WT and MLL+/- mice are only evident when they receive glucose overload. How do authors explain the same basal glucose blood levels (before injections)?
Authors theorize about downregulation of GLUT1/GLUT2 receptors, if so, what impact did the lack of expression in the liver, intestine, kidney, and central nervous system would have on the Knock-down mice? By the way, I do encourage to explore the expression of this protein in beta-cells.
Given these shortcomings the manuscript requires minor revisions.
Author Response
Thank you for reviewing our manuscript and for providing your valuable comments and suggestions. We appreciate your feedback and have responded to your comments.
1,Authors conclude that "MLL does not regulate tumorigenesis or apoptosis within islets", but it has been previously well-reported that MII deletion may induce leukemogenesis and fatal bone marrow failure occuring in a few weeks (in animal models). At what age were these KO mice sacrified? Did authors perform an assessment on haemathological features? Do authors consider the long-life period in mice before being sacrified enough to assure "MLL does not regulate tumorigenesis within islets"? It would be convenient a further explanation on histologycal analysis since refrence (44) do not fully explain this point.
Answer
All animal experiments were performed using 20-week-old mice. This information has been added to the Materials and Methods section.
Unfortunately, we did not conduct hematological examination. Reference citation 15 suggests that MLL is vital for maintaining an adequate number of hematopoietic progenitors and ensuring their appropriate differentiation, particularly in the myeloid and macrophage lineages. Reference citation 16 indicates a significant decrease in hematopoietic precursors in MLL mutant mice. This suggests that MLL can regulate hematopoietic precursor growth.
Indeed, we did not conduct confirmations at weekly intervals to definitively conclude that MLL is not involved in tumorigenesis. We have added the phrase " 20-week-old” in the Results and Discussion section. Unfortunately, we cannot illustrate it as a figure; however, we have confirmed via macroscopic examination no changes were observed in the pancreas of five wild-type and heterozygous mice at 50 weeks.
2,Authors should also further discuss about results showed in Figure 3. How do they explain, the islets in heterozygous MLL-knockout mice begin to synthesize more insulin 30 minutes after exposure to i.p glucose? (and blood sugar levels decrease at a time until they reach the same blood levels than wild-type mice after 120 min).
Answer
Considering that we intraperitoneally administered glucose and insulin into mice in vivo, various factors may be involved in blood glucose regulation. Therefore, the blood insulin levels (B) in IPGTT reflect the insulin levels after glucose load, confirming insulin secretion rather than synthesis from the pancreas.
Furthermore, ITT suggests comparable insulin resistance between both mouse groups. Accordingly, we infer that early insulin secretion may decrease in MLL knockout mice in response to glucose load; however, insulin resistance does not occur, resulting in similar blood glucose levels at 120 min.
Additionally, the possibility of GLUT1/GLUT2 action in organs such as the liver cannot be ignored, as indicated below.
These findings and conclusions have been added to the Discussion section (Lines 197–208).
3.It is also difficult to understand why differences in glucose homeostasis in WT and MLL+/- mice are only evident when they receive glucose overload. How do authors explain the same basal glucose blood levels (before injections)?
Answer
In terms of insulin secretion, we hypothesize the presence of both basal and additional secretion, with each being possibly governed by distinct mechanisms. The similar fasting blood glucose levels between wild-type and MLL heterozygous mice suggests similar insulin resistance and no differences in basal insulin secretion. Furthermore, in the present study, we focused on the role of MLL in additional insulin secretion. Mechanisms other than those elucidated in this study may be involved in basal insulin secretion. In the future, the mechanisms underlying MLL in both basal and additional secretion should be explored.
These points have been added to the Discussion section (Lines 197–208).
4.Authors theorize about downregulation of GLUT1/GLUT2 receptors, if so, what impact did the lack of expression in the liver, intestine, kidney, and central nervous system would have on the Knock-down mice? By the way, I do encourage to explore the expression of this protein in beta-cells.
Answer
Based on you comment, I have additionally discussed the functions of GLUT1 and GLUT2 in other organs and their potential involvement (Lines 250–253)
We plan to analyze these proteins in future studies.
Reviewer 2 Report
Comments and Suggestions for Authors
Thank you very much for Opportunity to read this manuscript. It’s well written in scientific language.
There are no issues detected. In my opinion it’s very high class paper. I would like to cangratulate Authors.
my suggestions :
1. all abbreviatios please spell out
2. in my opinion there is a lack of sub chapter „clinical implications” in Discussion
3. Spell check
thank you very much .
Author Response
Thank you for reviewing our manuscript and providing your valuable comments and suggestions. We appreciate your feedback and have responded to your suggestions.
1, all abbreviatios please spell out
Answer.
Based on your comment, we have correctly spelled out all abbreviations.
2, in my opinion there is a lack of sub chapter „clinical implications” in Discussion
Answer.
We have added the clinical implications of our study findings to the Discussion section (Lines 276–282).
3. Spell check
Answer.
We have commissioned a specialized scientific paper editing service for English proofreading. They have conducted a thorough spell check. We have attaching the certificate of editing from the company.

Reviewer 3 Report
Comments and Suggestions for Authors
Overall, the paper makes a valuable contribution to the understanding of the molecular mechanisms governing pancreatic beta cell function and glucose homeostasis. It provides important insights into the role of MLL in regulating glucose-sensitive gene expression and insulin secretion, with potential implications for the development of novel therapeutic strategies for diabetes.
The results section presents the experimental findings in a clear and organized manner. It effectively communicates the key observations, including changes in gene expression and insulin secretion in response to glucose stimulation and MLL manipulation.
The discussion section interprets the results in the context of the broader research field and provides insights into the mechanisms underlying glucose-sensitive gene expression and insulin secretion regulated by MLL in pancreatic beta cells. It highlights the significance of the findings, identifies potential limitations of the study, and proposes future directions for research.
Author Response
Answer
Thank you for reviewing our manuscript and providing your insightful comments. We appreciate your feedback and will continuously improve in our future research endeavors.